# YPEL3 Negatively Regulates Endometrial Function via the Wnt/β-Catenin Pathways during Early Pregnancy in Goats

**DOI:** 10.3390/ani12212973

**Published:** 2022-10-28

**Authors:** Jianguo Liu, Rendong Qiu, Ran Liu, Pengjie Song, Pengfei Lin, Huatao Chen, Dong Zhou, Aihua Wang, Yaping Jin

**Affiliations:** Key Laboratory of Animal Biotechnology of the Ministry of Agriculture, College of Veterinary Medicine, Northwest A&F University, Yangling 712100, China

**Keywords:** YPEL3, hormone, goat endometrial epithelial cells, Wnt/β-catenin, endometrial function

## Abstract

**Simple Summary:**

YPEL3 plays an important role in epithelial-mesenchymal transition (EMT) progression, however, its ability to regulate endometrial cell function and EMT during embryo implantation in ruminants remains unclear. In this study, we explored the molecular mechanism by which YPEL3 regulates endometrial function in goats. We found that YPEL3 acts as a negative regulator of endometrial function via the Wnt/β-catenin signaling pathways. These findings may provide a reference for improving conception rates in ruminants.

**Abstract:**

In ruminants, the establishment of pregnancy requires a series of structural and functional changes in the endometrium under the action of hormones, thereby providing an optimal environment for the implantation of the embryo. In this study, we explored the molecular mechanism by which YPEL3 regulates endometrial function during gestation in goats. We found YPEL3 expression was significantly downregulated during early gestation and that YPEL3 overexpression inhibited the expression of *ISG15*, but had no significant effects on the expression of *RSAD2* and *CXCL10* in goat endometrial epithelial cells (gEECs). In addition, YPEL3 silencing significantly inhibited PGF_2α_ secretion and the expression of the prostaglandin synthesis-related rate-limiting enzyme-encoding genes *PGFS* and *PTGES*, with no significant effect on the expression of *PTGS1* and *PTGS2*. Moreover, YPEL3 inhibited the expression of vimentin and β-catenin and pretreatment of gEECs with the β-catenin activator CHIR99021 prevented a YPEL3-induced decrease in vimentin expression. Collectively, our findings confirm that, as a hormone-regulated factor, YPEL3 regulates endometrial function by inhibiting the Wnt/β-catenin signaling pathway and provide new insights for further clarification of the mechanism by which YPEL3 functions during early pregnancy in ruminants.

## 1. Introduction

In the process of establishing pregnancy in ruminants, it is not only necessary for the embryo to grow, elongate, and finally adhere to the endometrium, but also for the structure and function of the endometrium to evolve under the influence of various hormones, transcription factors and cytokines, so as to provide the best environment for the implantation of the embryo [1,2]. This process is known as the establishment of receptivity [3]. The close relationship between a well-developed embryo and a highly receptive endometrium is the key to successful embryo implantation [4]. Estrogen (E_2_), progesterone (P_4_) and interferon-tau (IFN-τ) activate and maintain endometrial remodeling, uterine growth and endometrial gland morphogenesis and secretion function in ruminants during early pregnancy [5,6].

IFN-τ is a type I interferon that has been identified as a pregnancy recognition signal in ruminants, such as cattle, sheep and goats. IFN-τ binds to the interferon-α/β receptor (IFNAR) to activate the expression of a series of classical or non-classical interferon-stimulated genes (ISGs) [7]. Ubiquitin cross-reactive protein (ISG15), which is encoded by a classical ISG, mediates post-translational modification of proteins, and plays a crucial role in regulating trophectoderm cells proliferation, embryo development and blastocyst incubation during early pregnancy in ruminants [8]. Chemokine (C-X-C motif) ligand 10 (CXCL10) promotes gestational body elongation in early pregnancy and enhances adhesion between the uterus and the embryonic trophoblast on day 17 of gestation [9]. S-adenosyl domain-containing protein 2 (RSAD2) mainly regulates uterine receptivity to embryo implantation by inducing uterine antiviral activity and regulating immune cell function [10]. In sheep and cattle, the endometrium in early pregnancy increases the production of prostaglandins (PGs), such as PGE_2_, PGF_2α_, PGI_2_ and PGD_2_ [11]. In the PG production pathways, prostaglandin-endoperoxide synthase 1 (PTGS1) and prostaglandin-endoperoxide synthase 2 (PTGS2) catalyze the conversion of arachidonic acid (AA) into PGH_2_, which is subsequently converted into PGF_2α_ and PGE_2_ by prostaglandin F synthase (PGFS) and prostaglandin E synthase (PTGES), respectively [12]. Studies have also shown that PGs can mediate the effects of P_4_ and IFN-τ on the endometrium and promote embryo elongation [13]; however, the specific regulatory mechanism remains to be fully elucidated. 

After the blastocyst adheres to the uterus, trophoblast cells undergo epithelial-mesenchymal transition (EMT) and invade the endometrial stromal layer, transforming from an epithelioid phenotype to a mesenchymal phenotype with effective adhesion and invasion capabilities [3]. As the trophectoderm undergoes EMT, the endometrial epithelium also undergoes EMT-related changes [14]. The EMT process is also accompanied by the increased expression of mesenchymal markers, such as vimentin and N-cadherin, and the loss of epithelial markers, such as E-cadherin, tight junction protein 1 and plaque globin [15,16]. EMT is crucial for embryo implantation, and uncontrolled EMT may lead to the development of pregnancy complications [17]. Therefore, the timing and extent of EMT must be strictly controlled to achieve normal and effective implantation.

The *Yippee* gene was first discovered through investigations of a Drosophila protein that interacts with heme, a member of the immunoglobulin superfamily [18]. Subsequently, five highly homologous Yippee-like (*YPEL*) genes, including *YPEL3*, were identified in humans and mice [19]. YPEL3 inhibits the EMT process in cancer cells. In nasopharyngeal cancer cells, YPEL3 promotes E-cadherin expression by inhibiting the entry of β-catenin into the nucleus, and inhibits vimentin expression to regulate the EMT process, YPEL3 also inhibits cell proliferation and migration [20]. In human colon cancer tissues, YPEL3 also suppresses cancer cell proliferation, migration and invasion by inhibiting the Wnt/β-catenin signaling pathway, which controls many biological phenomena, such as fat metabolism and animal development [21,22,23]. Thus, YPEL3 has been shown to play an important role in EMT progression, although its ability to regulate endometrial cell function and EMT during embryo implantation remains unclear. Based on previous studies, we hypothesized that YPEL3 is involved in the regulation of endometrial function during embryo implantation in goats. Therefore, the objective of this study was to explore the regulation of YPEL3 endometrial expression during early gestation in goats as well as its role in establishing endometrial receptivity. These findings provide new insights that can be used to further clarify the mechanism of embryo implantation in ruminant livestock and improve the pregnancy rate of livestock embryos. 

## 2. Materials and Methods

### 2.1. Tissue Collection

Dairy goats were provided by the Experimental Animal Center of Northwest A&F University, Yangling, China. The uteruses were collected on pregnancy days 5 (P5, n = 3), 15 (P15, n = 3), and 18 (P18, n = 3) and immediately fixed in 4% paraformaldehyde for later use. All experimental procedures were performed in accordance with the Committee for the Ethics on Animal Care and Experiments of Northwest A&F University (China).

### 2.2. Cell Culture and Drug Treatment

The gEECs were removed from liquid nitrogen and cultured in DMEM/F-12 (HyClone, SH30023.01) containing 10% fetal bovine serum (FBS, ZETA, Z7186FBS-500) at 37 °C in a humidified incubator under 5% CO_2_. The medium was changed every two days. After 2–3 consecutive passages, the cells were seeded into a 6-well plate (4 × 10^5^ cells per well). When the cells reached approximately 70% confluence, the medium was replaced with DMEM/F-12 containing 0.1% bovine serum albumin (BSA, Sigma, A1933). After 24 h, P_4_ (10^−7^ M) and E_2_ (10^−9^ M) were added, and the cells were incubated for 12 h before being cultured with 20 ng/mL IFN-τ for a further 12 h.

### 2.3. Immunohistochemistry

Goat uterine tissues were fixed with 4% paraformaldehyde for 24 h and dehydrated through a graded series of ethanol solutions. Sections (thickness, 7 μm) were mounted onto glass slides precoated with poly-L-lysine solution. The dehydrated sections were placed in citrate buffer (PH 6.0) for antigen retrieval using the microwave heating method. After washing in phosphate-buffered saline (PBS), the slides were placed in 0.3% hydrogen peroxide/methanol for 1 h to inactivate endogenous peroxidase. The sections were then incubated in 10% pre-immune serum (Maixin-Biotech, Fuzhou, China) for 30 min at 37 °C prior to incubation with anti-YPEL3 antibody (1:100, ABclonal Technology, Wuhan, China) for 12 h at 4 °C. The negative control sections were incubated overnight with pre-immune serum. After washing, the sections were incubated with biotinylated anti-rabbit IgG antibody (Maixin-Biotech, Fuzhou, China) for 30 min at 37 °C, washed three times with PBS and then incubated with horseradish-peroxidase-labeled streptavidin for 30 min at 37 °C. Finally, after staining with DAB chromogenic solution, the sections were valuated under a microscope (Nikon, Tokyo, Japan). All antibodies were diluted with PBS.

### 2.4. Real-time Quantitative PCR

Total cellular RNA was extracted with TRIzol (TaKaRa, Tokyo, Japan) and reverse transcribed into cDNA using the Evo M-MLV RT reagent kit (AG Bio, Changsha, China). A sample of the cDNA (20 ng) was used as a template to analyze mRNA levels by quantitative real-time PCR (qPCR) with the SYBR qPCR Master Mix kit (Vazyme Bio, Najing, China) using the Bio-Rad CFX96 system (Bio-Rad, Hercules, CA, USA) according to the manufacturer’s protocol. The primer sequences are listed in Appendix A. Expression levels were analyzed by the 2^−ΔΔCt^ method. The expression of mRNA was normalized by the *GAPDH* gene and served as the control gene in all samples.

### 2.5. Western Bolt Analysis

Western blot analysis was conducted according to previously described procedures [24] with the following antibodies: anti-YPEL3 (1:500, ABclonal Technology, Wuhan, China), anti-N-cadherin (1:1000, Abcam, Cambridge, UK), anti-vimentin (1:500, Santa Cruz, California, USA), anti-β-catenin (1:500, Santa Cruz, California, USA), anti-β-actin (1:5000, Sanjian Biotech, Wuhan, China). Binding of antibodies was detected using HRP-labeled goat anti-rabbit or anti-mouse immunoglobulin (1:5000, Wuhan, Sanjian Biotech). Subsequently, the proteins were visualized using the Gel Image System (Tannon Biotech, Shanghai, China). The bands were quantified by densitometry using Imagine-Pro Plus (Media Cybernetics, Rockville, MD, USA). All antibodies were diluted with Antibody Diluent (NCM Biotech, WB500D).

### 2.6. Immunofluorescent Staining

Cell sections were seeded in 24-well cell culture dishes, and then processed for immunofluorescent staining when the cells reached approximately 60–70% confluence. After the culture medium was discarded, the cells were washed twice with pre-cooled PBS, and then immersed in 300 μL of 4% paraformaldehyde for 30 min at room temperature. The cells were then washed with PBS (5 min per wash). Subsequently, the cells were permeabilized by the addition of 300 μL of 0.1% Triton X-100 PBST and then blocked with 300 μL of PBS containing 1% BSA for 1 h at room temperature. Finally, the cells were incubated with anti-YPEL3 (1:200, ABclonal Technology, Wuhan, China) primary antibody followed by a fluorescent secondary antibody (1:800, Invitrogen, Waltham, MA, USA). The cells were fixed on a glass slide and observed by laser-scanning confocal microscopy (Nikon Inc., Melville, NY, USA). All antibodies were diluted with PBS.

### 2.7. Cell Transfection with Interference Target Sequence

Recombinant lentiviral vectors encoding the YPEL3 short hairpin RNA (shYPEL3) and negative control shRNA (shN) (Appendix A) were designed by BLOCK-iT™ RNAi designer online software. Virus packaging and cell transfection were performed as described previously [25,26]. Specifically, recombinant lentiviral vectors were packaged and transduced into HEK 293T cells. After 48 h of continuous culture, the cell supernatant was collected, centrifuged at 2000 r/min for 10 min, then filtered and stored at −80 °C for later use. 3 mL of this virus stock solution collected above was added to 50% density gEECs, and 8μg/μL of polybrene (Sigma, 107689) was added at the same time. After 12 h, the mixture was replaced with normal culture medium, and after 48 h, purine (Sigma, P55805) was added to screen for 4–5 days to select a stable cell line.

### 2.8. Enzyme Linked Immunosorbent Essay (ELISA)

The cells were seeded into 6-well plates and cultured to >90% confluence. The culture supernatant was then collected and the levels of PGF_2α_ were analyzed using a commercial ELISA kit (JYM, JYM0120Go, China) according to the manufacturer’s instructions. Specifically, blank wells, standard wells and sample wells were set on the microtiter plate, different concentrations of standards were added to the standard wells, 50 μL of supernatant was added to the sample wells and incubated in a 37 °C incubator for 1 h. Next, except for the blank wells, we added 100 μL of secondary antibody to each well, sealed the plate with a membrane, and incubated it in a 37 °C incubator for 1 h. Furthermore, we discarded the liquid, patted the ELISA plate dry on absorbent paper, added 350 μL of washing solution to each well, let them stand for 1 min, and repeated the operation 5 times. Subsequently, substrate A and Substrate B were mixed at a ratio of 1:1, and 100 μL was added to each well, and incubated at 37 °C for 15 min in the dark. Finally, 50 μL of stop solution was added to each well, and the OD of each well was measured immediately at 450 nm using a multifunctional enzyme labeling instrument (BioTek, Cranbury, NJ, USA). The concentration of PGF_2α_ in the sample was calculated according to the standard curve.

### 2.9. Statistical Analysis

Data were reported as the arithmetic means ± standard error of the mean (S. E. M) of three samples. One-way ANOVA followed by Tukey’s post-hoc test and Fisher’s least significant difference test were used for multiple comparisons. *p* < 0.05 was set as the threshold for statistical significance.

## 3. Results

### 3.1. YPEL3 Expression Was Downregulated in Early Pregnancy

To explore the role of YPEL3 in goat embryo implantation, we performed immunohistochemical staining to detect the expression and localization of YPEL3 in the endometrium on days 5 (P5), 15 (P15) and 18 (P18) of gestation. As shown in Figure 1A, YPEL3 was localized mainly in the luminal and glandular epithelial cells of the endometrium, with a small amount detected in the stromal cells. YPEL3 protein expression was significantly reduced at P18 compared to the levels detected at P5 and P15. Furthermore, we treated gEECs with E_2_ and P_4_ or combined with IFN-τ to simulate the hormonal environment of goat embryo implantation in early pregnancy. We found that there were no significant differences in the *YPEL3* mRNA levels (Figure 1B), between the E_2_ + P_4_ and control groups, while the expression levels in the E_2_ + P_4_ + IFN-τ group were significantly decreased. At the protein level (Figure 1C), YPEL3 expression was also significantly inhibited in the E_2_ + P_4_ + IFN-τ group compared with that in the control group. These results indicated that YPEL3 expression was significantly downregulated in the endometrium of goats during early pregnancy.

### 3.2. YPEL3 Expression in gEECs Was Significantly Affected by YPEL3 Silencing or Overexpression 

To assess the role of YPEL3 in regulating endometrial function during early pregnancy, we constructed pcDNA3.1-YPEL3 and shYPEL3 vectors to induce YPEL3 overexpression and silencing, respectively, in transfected gEECs. Following transfection of gEECs with pcDNA3.1-YPEL3 for 48 h, YPEL3 expression was significantly upregulated at both the mRNA (Figure 2A) and protein (Figure 2B) levels. In contrast, following the transfection of gEECs with the pCD513B-U6-shYPEL3 lentivirus, real-time quantitative PCR analysis showed that the efficiency of interference by shYPEL3-1 reached 80% (Figure 2C). Furthermore, Western blot (Figure 2D) and immunofluorescence (Figure 2E) analyses confirmed a significantly decreased YPEL3 protein expression following transfection with the shYPEL3 vector. These data indicated that pcDNA3.1-YPEL3 and pCD513B-U6-shYPEL3 lentivirus efficiently affected the expression of YPEL3 at both the transcriptional and translational levels.

### 3.3. YPEL3 Affected Endometrial Receptivity and the Secretion of PGF_2α_

To investigate the ability of YPEL3 to affect the expression of progestational elongation genes and thus, endometrial receptivity during embryo implantation, YPEL3-overexpressing gEECs were treated with E_2_, P_4_ and IFN-τ, and the expression of *ISG15*, *RSAD2* and *CXCL10* was detected. We found that YPEL3 overexpression inhibited the expression of *ISG15* (Figure 3A) but had no significant effect on the expression of *RSAD2* (Figure 3B) and *CXCL10* (Figure 3C). We also explored the effect of YPEL3 on prostaglandin synthesis in gEECs. Transfection of gEECs with shYPEL3 significantly inhibited PGF_2α_ secretion (Figure 3D) and the expression of the prostaglandin synthesis-related rate-limiting enzyme genes *PGFS* and *PTGES* (Figure 3E) but had no significant effect on *PTGS1* and *PTGS2* expression. These data suggested that YPEL3 controls endometrial development during implantation by regulating *ISG15* expression and the secretion of PGF_2α_.

### 3.4. YPEL3 Significantly Affected the Expression of EMT Marker Proteins

We next aimed to verify the ability of YPEL3 to affect the expression of EMT marker proteins through hormonal regulation by treating gEECs with E_2_, P_4_ and IFN-τ prior to the evaluation of the expression of the EMT marker proteins vimentin and N-cadherin in the hormonal model of early pregnancy. Compared with the control group, the expression of vimentin and N-cadherin were significantly increased in the E_2_ + P_4_ group and the E_2_ + P_4_ + IFN-τ groups, indicating that gEEC EMT can occur in early pregnancy hormone models (Figure 4A). In addition, Western blot analysis showed that YPEL3 inhibited the expression of vimentin but had no significant effect on N-cadherin (Figure 4B). In contrast, YPEL3 silencing was shown to promote the expression of vimentin but had no significant effect on the expression of N-cadherin (Figure 4C). These results indicated that YPEL3 inhibits the expression of vimentin in gEECs.

### 3.5. YPEL3 Was Involved in Regulation of the Wnt/β-Catenin Signaling Pathway

To verify the ability of YPEL3 to regulate the expression of EMT marker proteins in gEECs via the β-catenin signaling pathway, we treated gEECs with overexpression or silencing of YPEL3 with E_2_, P_4_ and IFN-τ prior to the analysis of β-catenin expression. Western blot analysis showed that YPEL3 overexpression inhibited the expression of β-catenin (Figure 5A), while its expression was promoted by YPEL3 silencing (Figure 5B). Immunofluorescence staining showed that YPEL3 overexpression reduced β-catenin expression in the cytoplasm and its accumulation in the nucleus (Figure 5C), while the opposite pattern of β-catenin expression was observed after YPEL3 knock-down (Figure 5D). This suggested that YPEL3 negatively regulates the Wnt/β-catenin signaling pathway.

### 3.6. YPEL3 Regulated the Mesenchymal State of gEECs via the Wnt/β-Catenin Signaling Pathway

To further verify the ability of YPEL3 to regulate the mesenchymal state of gEECs via the Wnt/β-catenin signaling pathway, we analyzed the expression of EMT marker proteins in gEECs overexpressing YPEL3 which were then treated with the β-catenin activator CHIR99021. As shown in Figure 6, the protein expression of both vimentin and β-catenin was inhibited after YPEL3 overexpression, although the effect on the latter was not statistically significant. However, after the addition of the β-catenin activator CHIR99021, the protein levels of β-catenin increased, and the decrease in vimentin expression caused by YPEL3 overexpression was ameliorated. These data indicated that YPEL3 regulates the mesenchymal state of gEECs via the Wnt/β-catenin signaling pathway.

## 4. Discussion

Endometrial receptivity (the engraftment window) is a limited period of early pregnancy in which biological processes occur at the molecular and cellular levels to promote embryo engraftment and establish pregnancy [27]. During the development of receptivity, the secretory function of the endometrium changes, and EMT occurs to facilitate the adhesion and invasion of the embryo [14,28]. Transcriptomic sequencing of goat endometrium at different stages of embryo implantation showed that the expression of YPEL3 was inhibited [29]. YPEL3 inhibits the entry of β-catenin into the nucleus to regulate the process of EMT [30]. Hence, in the present study we explored the molecular mechanism by which YPEL3 regulates endometrial function during gestation in goats to provide a new therapeutic target for improving conception rates in ruminants. 

During embryo implantation, the endometrium undergoes dynamic changes that are controlled by hormones to provide the most suitable environment for blastocyst development. The time of embryo adhesion varies in different species. In goats, the embryo adheres firmly to the endometrium on day 18 of pregnancy [31]. In this study, we showed that YPEL3 protein expression was significantly reduced on day 18 of pregnancy compared to that at day 5 and day 15, suggesting that YPEL3 is negatively regulated during goat embryo implantation. In our previous studies, we showed that treatment of gEECs with a combination of E_2_, P_4_ and IFN-τ simulated the hormonal environment of the endometrium during embryonic adhesion, thus promoting gEEC receptivity [24]. Using this model, we also confirmed that YPEL3 expression was downregulated at the early stages of embryo implantation, which was consistent with the results of the immunohistochemical analysis. Hence, we speculated that YPEL3 regulates various biological processes, such as EMT and the receptivity of the endometrium, during goat embryo implantation. 

During early pregnancy in sheep and cattle, prostaglandins are synthesized by the conceptus and endometrium to promote embryo implantation [11]. The treatment of gEECs with a combination of E_2_, P_4_ and IFN-τ can alter the expression of *PTGS1*, *PTGS2*, *PGFS* and *PTGES* and increase the ratio of PGE_2_:PGF_2α_, thereby helping to protect the corpus luteum and maintain pregnancy [26]. Therefore, having shown that YPEL3 has a negative regulatory effect on endometrial function during embryo implantation in goats, we next investigated the specific mechanism underlying the influence of YPEL3 on endometrial function and the effects of YPEL3 on prostaglandin secretion, progestational elongation gene expression and EMT in gEECs. We found that YPEL3 silencing significantly inhibited the expression of *PGFS* and *PTGES* as well as the secretion of PGF_2α_. Therefore, we speculate that YPEL3 inhibits the secretion of PGF_2α_ by negatively regulating the expression of PGFS and increasing the ratio of PGE_2_:PGF_2α_ to maintain pregnancy. This information offers a new insight into the changes in prostaglandin secretion during goat embryo implantation, although the specific regulatory mechanism remains to be fully elucidated. 

IFN-τ, which functions as a pregnancy recognition signal in ruminants, regulates the expression of some canonical ISGs (such as *ISG15*, *RSAD2* and *CXCL10*) and non-canonical ISGs to change the intrauterine environment and facilitate embryo implantation [8,32,33]. ISG15 is the main regulator of trophectoderm cells proliferation, embryo development and blastocyst formation in ruminants during early gestation [34]. YPEL3 expression resulted in *ISG15* suppression, while the expression of *RSAD2* and *CXCL10* was not significantly affected. Therefore, we hypothesize that YPEL3 controls endometrial development during implantation by regulating the expression of ISG15.

YPEL3 is a tumor suppressor that can inhibit the EMT process in a variety of tumor cells [21,30]. Following implantation of the embryo into the endometrium, the endometrial epithelial cells undergo EMT. This process is manifested by a change in cell polarity, remodeling of the cytoskeleton, the loss of epithelial markers, increased mesenchymal markers expression (vimentin and N-cadherin, etc.) and the gain of migration potential, all changes that are conducive to the invasion of trophoblastic cells [35,36]. In this study, we showed that YPEL3 inhibited vimentin expression in gEECs, but had no significant effect on the expression of N-cadherin, suggesting that YPEL3 affects endometrial function by inhibiting the EMT process.

The level of β-catenin in the endometrium is critical for successful pregnancy. During mouse embryo implantation, excessively high or low β-catenin levels may lead to implantation failure [37]. Abnormalities in the Wnt/β-catenin signaling pathway in human placental tissue can also lead to early pregnancy loss [38]. In this study, we found that YPEL3 inhibited the expression of β-catenin in gEECs and affected the nuclear entry of YPEL3. Treatment with the β-catenin activator CHIR99021, ameliorated the reduction in vimentin expression levels caused in gEECs by overexpression of YPEL3. Stabilization of β-catenin is essential for Wnt/β-catenin signaling and studies have shown that multi-drug resistance protein 4 (MRP4) can enhance the stability of β-catenin through protein–protein interaction, thus promoting the implantation of mouse embryos [39]. Although YPEL3 can also negatively regulate the expression of vimentin in gEECs via the Wnt/β-catenin signaling pathway to obtain migration characteristics, there seems to be no protein–protein interaction between YPEL3 and β-catenin. However, it is not yet clear whether this regulation involves the destabilization of β-catenin or another mechanism.

## 5. Conclusions

In this study, we showed that YPEL3 controls goat embryo implantation by regulating vimentin and ISG15 expression and PGF_2α_ secretion in gEECs during the implantation stage. Thus, we have provided evidence that, as a hormone-regulated factor, YPEL3 regulates endometrial function by inhibiting the Wnt/β-catenin signaling pathway. Although further in vivo verification is required, these findings may provide a reference for improving conception rates in ruminants.

## Figures and Tables

**Figure 1 animals-12-02973-f001:**
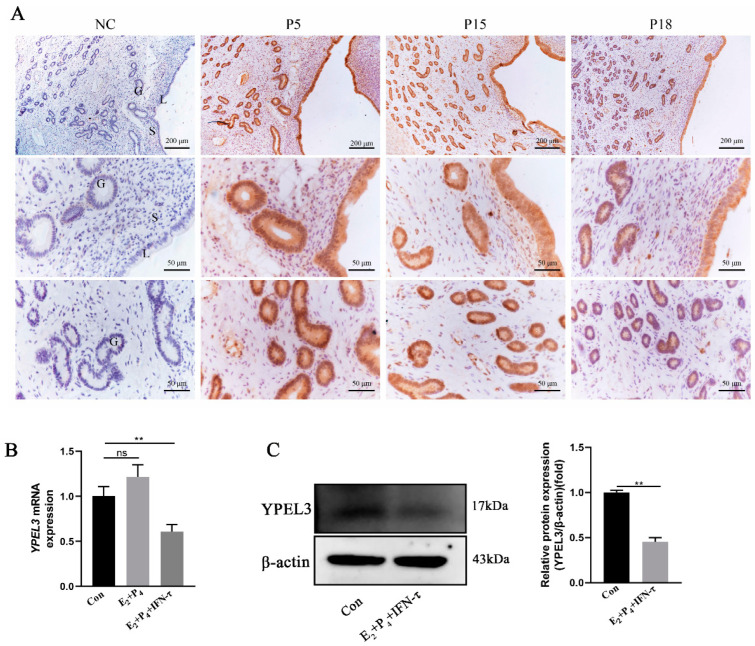
YPEL3 expression in the goat endometrium during early pregnancy and hormone treatment. (**A**) Immunohistochemical staining of YPEL3 protein in goat uterine tissues during early pregnancy (days 5, 15 and 18 of the pregnancy). (**B**) Real-time quantitative PCR analysis of *YPEL3* mRNA levels after E_2_, P_4_ and IFN-τ treatment. (**C**) Western blot analysis of YPEL3 protein expression following E_2_, P_4_ and IFN-τ treatment. L, luminal epithelium; G, glandular epithelium; S, stromal cells. Scale bars = 200 μm or scale bars = 50 μm. ** *p* < 0.01, ns, not significant (*p* > 0.05).

**Figure 2 animals-12-02973-f002:**
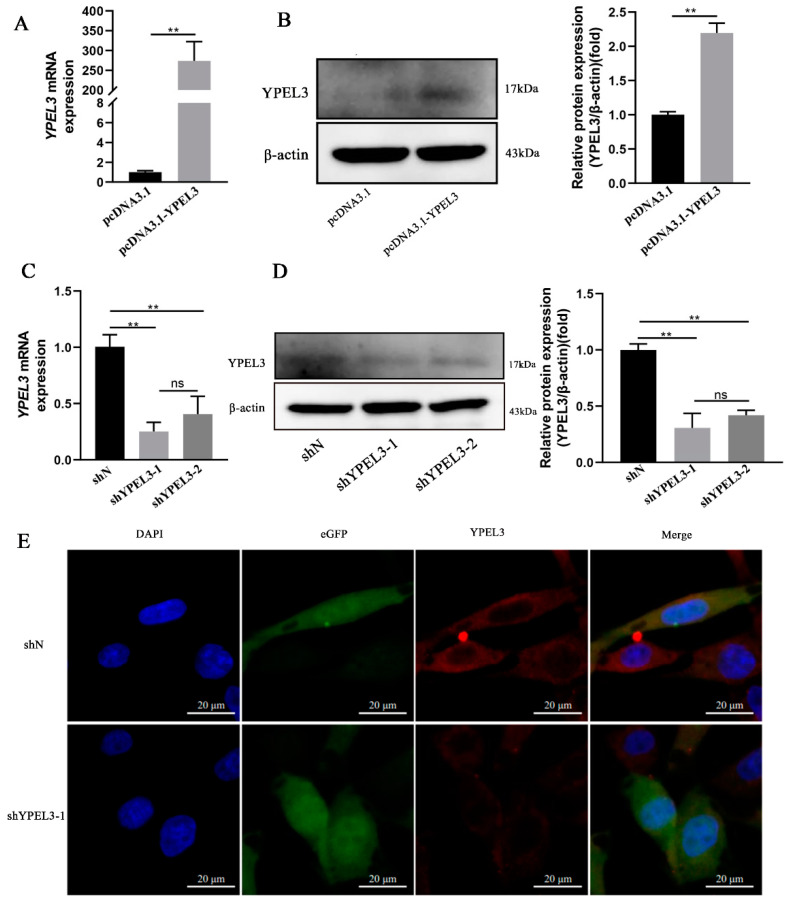
Confirmation of silencing and overexpression of YPEL3 in gEECs. (**A**) Real-time quantitative PCR analysis of *YPEL3* mRNA overexpression efficiency in gEECs transfected with pcDNA3.1 and pcDNA3.1-YPEL3 for 48 h. (**B**) Western blot analysis of YPEL3 protein overexpression efficiency in gEECs transfected with pcDNA3.1 and pcDNA3.1-YPEL3 for 48 h. (**C**) Real-time quantitative PCR analysis of *YPEL3* mRNA levels in gEECs transfected with shN and shYPEL3 for 48 h. (**D**) Western blot analysis of YPEL3 protein overexpression efficiency in gEECs transfected with shN and shYPEL3 for 48 h. (**E**) Confocal microscope images of YPEL3 expression in gEECs transfected with shN and shYPEL3 for 48 h. Scale bars = 20 μm. ** *p* < 0.01, ns, not significant (*p* > 0.05).

**Figure 3 animals-12-02973-f003:**
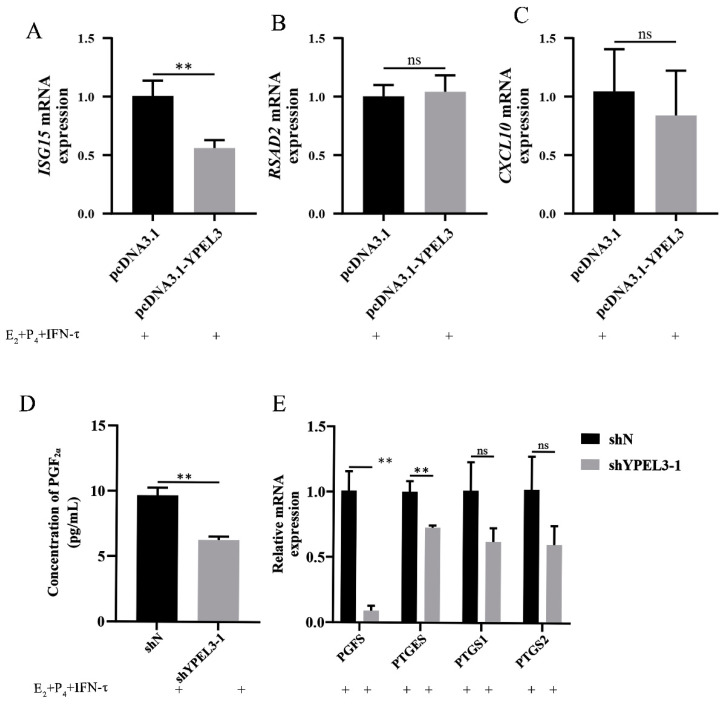
YPEL3 altered endometrial receptivity and the secretion of PGF_2α_. (**A**–**C**) Real-time quantitative PCR analysis of the expression of genes (*ISG15*, *RSAD2*, *CXCL10*) associated with promoting conceptus elongation in gEECs overexpressing YPEL3 and treated with E_2_, P_4_ and IFN-τ. (**D**) ELISA analysis of PGF_2α_ secretion by gEECs with YEPL3 silencing and treated with E_2_, P_4_ and IFN-τ. (**E**) Real-time quantitative PCR analysis of the expression of genes encoding rate-limiting PG synthesis enzymes (*PGFS*, *PTGES*, *PTGS1*, *PTGS2*) in gEECs with YEPL3 silencing and treated with E_2_, P_4_ and IFN-τ. ** *p* < 0.01, ns, not significant (*p* > 0.05).

**Figure 4 animals-12-02973-f004:**
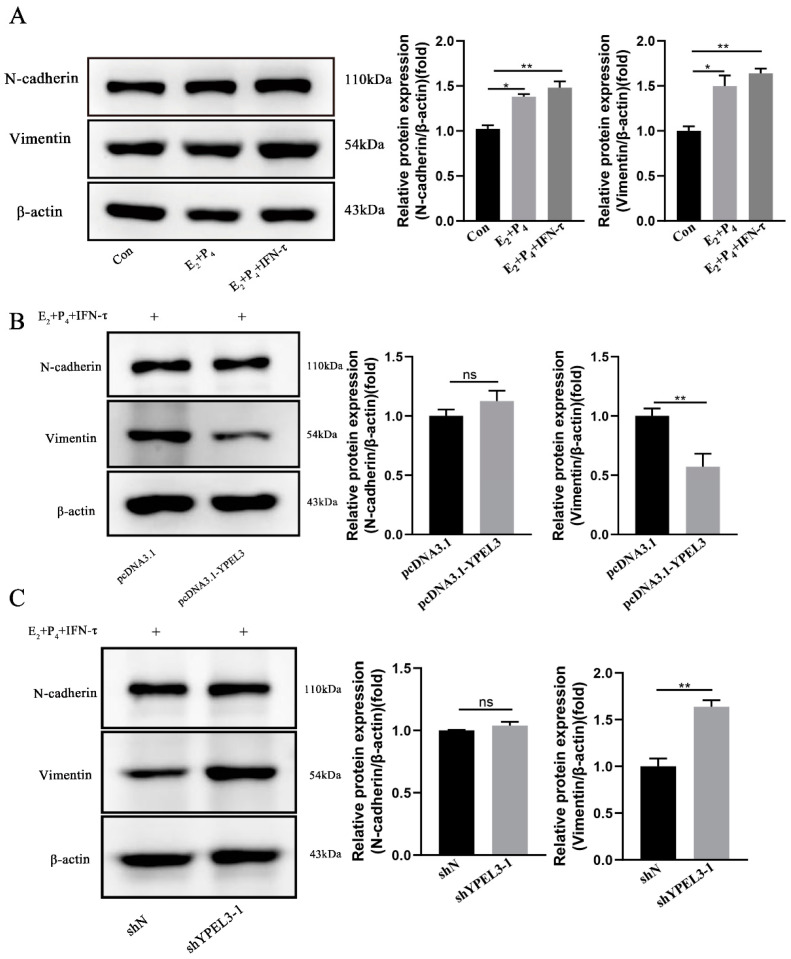
The effect of YPEL3 on EMT marker expression in gEECs. (**A**) Western blot analysis of the EMT marker proteins N-cadherin and vimentin after treatment with or without E_2_, P_4_ and IFN-τ. Western blot analysis of the EMT marker protein expression in gEECs with silencing (**B**) and overexpression (**C**) of YPEL3 and treated with E_2_, P_4_ and IFN-τ. * *p* < 0.05, ** *p* < 0.01, ns, not significant (*p* > 0.05).

**Figure 5 animals-12-02973-f005:**
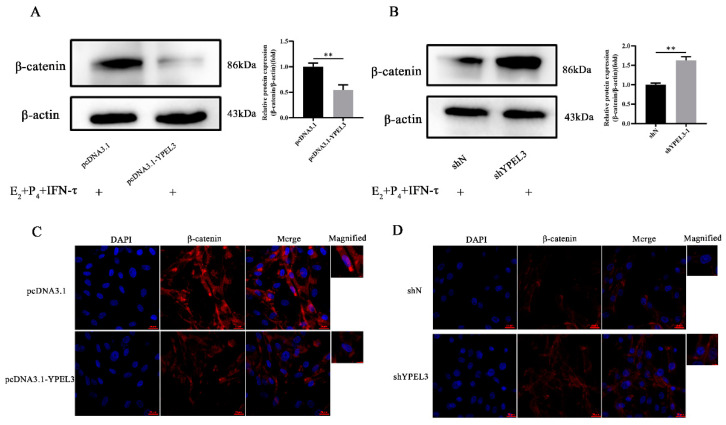
YPEL3 regulated the Wnt/β-catenin signaling pathway. Western blot analysis of the expression of β-catenin in gEECs with YEPL3 silencing (**A**) and YEPL3 overexpression (**B**) and treated with E_2_, P_4_ and IFN-τ. Confocal microscope images of immunohistochemical staining of β-catenin expression and nuclear translocation in gEECs with YEPL3 overexpression (**C**) or YEPL3 silencing (**D**) and treated with E_2_, P_4_ and IFN-τ. ** *p* < 0.01.

**Figure 6 animals-12-02973-f006:**
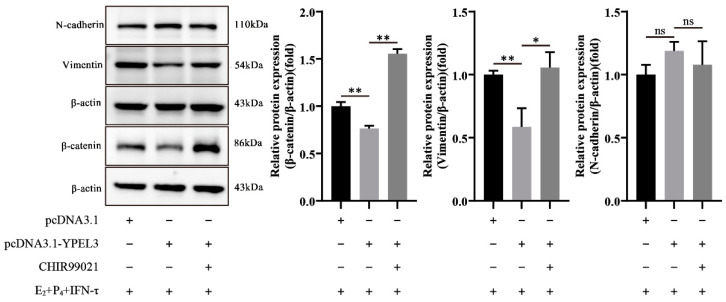
The effect of β-catenin activation on gEEC EMT. Western blot analysis of the expression of EMT markers (N-cadherin, vimentin) and β-catenin proteins in gEECs pre-treated with 5 μM CHIR99021 before E_2_, P_4_ and IFN-τ treatment. * *p* < 0.05, ** *p* < 0.01, ns, not significant (*p* > 0.05).

## Data Availability

The data presented in this study are available upon request to the corresponding author.

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
