# Peer review of "YPEL3 Negatively Regulates Endometrial Function via the Wnt/β-Catenin Pathways during Early Pregnancy in Goats"

_animals, 2022, doi:10.3390/ani12212973_

Round 1

Reviewer 1 Report

The manuscript entitled "YPEL3 negatively regulates endometrial function via Wntβ-catenin pathways during early pregnancy in goats" by Liu et al. to evaluate the effect of YPEL3 regulates the endometrial function via Wnt/β-catenin using overexpression and gene knockdown. The study has some interesting find, however, the experiment has many questions.

1、 There are many format errors in all manuscript, such as, line 91 4×105 cells per well, line 92 0.1% BSA, line 113 2-ΔΔCt, line 103, 105, 106, inconsistent for °C. please check the format in all manuscript.

2、 Introduction: The author should add a hypothesis by previous study conclusion, then, the objective of this study was to …….

3、 Line 141 P < 0.05 should be P

4、 The quality of Western blotting bands for YPEL3 is weak in all manuscript, and it is recommended to redo Western blotting for YPEL3.

5、 Line 134 The gEECs infected by lentivirus need to be descripted clearly.

6、 Western blotting bands for YPEL3 has no profound alter in shN, shYPEL3-1, and shYPEL3-2 from Figure 2D, however, relative protein expression has significant difference. Please explain it.

7、 How to detect the PGF2α secretion by gEECs? Which is company PGF2α Kit? Please add details.

8、 Compared with Con, Western blotting bands for N-cadherin and vimentin have no profound change in E2 and P4 from Figure 4A, but, the author think that N-cadherin and vimentin relative protein expression has significant difference. Please explain it.

Reviewer 2 Report

The objective of this study was to investigate the effect of YPEL3 on endometrital function. The study is well presented and it tells a story. These findings may provide a new therapeutic target for improving conception rates in ruminants. It is suggested to publish after revision. However, there are some details to notice here.

 1.What cell type was utilized for RNA extraction

 2.How much RNA was used in PCR? Was it one-step or 2-step PCR?

 3. I suggest the cite following paper in introduction part For more information you can read below reference

-RNA-Seq reveals the potential molecular mechanisms of bovine KLF6 gene in the regulation of adipogenesis. International Journal of Biological Macromolecules, 195, 198-206.

4. My suggestion is the section 2.2 should contain sufficient information to allow an independent researcher to repeat your experiments.

5. I also suggest state the dilution factors for each antibody in the full text.

6. Please revise the superscript and subscript problems in the full text writing.

7. Change the first part of the discussion, the discussion should start with a paragraph reminding the reader of the aim of your not repeated for the background information.

In conclusion, the research presented is interesting, well planned and carried out. The manuscript can still be improve revise. Nevertheless, I believe that this work deserves publication in after the inclusion of corrections.

Reviewer 3 Report

I reviewed the article titled: “YPEL3 negatively regulates endometrial function via Wnt/β-catenin pathways during early pregnancy in goats” and I found it well prepared. The authors presented the influence of YPEL3 protein on the regulation of the endometrial functions. In my opinion the article is suitable to be published in Animals upon reviewing the following points.

Simple Summary:

[72] All gene symbols should be written in italic

Materials and methods:

[84] The authors conducted the study on day 5, 15 and 18. Why these three periods were chosen? The number of repeats for each day is three, is it enough for the reliability of the results? On the other hand I suppose it is not so easy to obtain such tissue for the experiment.

[114] Based on what methodology the Authors choose the GAPDH gene for the analysis? Please explain.

[118] The cells were, not was.

[129] Subsequently, the cells (typo)

[132] The cells were was (typo)

Discussion:

[264] day 18 (typo)

Reviewer 4 Report

The authors Jianguo Liu et al. have submitted a manuscript in animals,      section: Animal Reproduction (ID: 1943313), entitled “YPEL3 negatively regulates endometrial function via Wnt/β-catenin pathways during early pregnancy in goats”.

The authors report a study on goat YPEL 3 gene, analyzed the transcript, protein and the effect on overexpression or silencing, in goat uterus during early pregnancy and on colture cells. We know that during embryo implantation numerus’s process are sequentially involving, as hormones, transcriptional factors and various structural proteins.

YPEL3 play a naturally endometrial function, and is regulate during goat embryogenesis. In fact, YPEL3 the transcript and protein level change during early gestation and is able to regulate genes as β-catenin essentially for normally life of embryo, influencing the expression of different genes.

I believe this study may be useful to increase the knowledge about the role of YPEL3 during pregnancy, a process involving most factor, actually not all characterized.

Finally, I consider the paper suitable for publication in animals as it is.

Round 2

Reviewer 1 Report

The revised manuscript is well done. However, the manuscript have still many format issues. Please check them carefully. The number and symbol should have a blank. Line 90. 5 (P5, n = 3), 15(P15, n = 3) are inconsistent.  

Author Response

Dear Reviewer:

Thank you very much for your very helpful advices on our manuscript (ID: animals-1943313). We have made changes with the manuscript according to your comments. And the detailed response is listed below.

A revised version has been uploaded to the website. Please feel free to let me know if you have any further suggestions. Thank you again for all your generous help.

Best Regards,

Comment and Response:

Point 1: The revised manuscript is well done. However, the manuscript have still many format issues. Please check them carefully. The number and symbol should have a blank. Line 90. 5 (P5, n = 3), 15(P15, n = 3) are inconsistent.

Response 1: Thank you very much for your kind suggestion, which improves our manuscript a lot. We have tried our best to make many changes concerning the manuscript format based on your comments as well as the template provided by the journal. Thank you again for your very helpful advices on the improvement of our manuscript.